# The Climate and Nutritional Impact of Beef in Different Dietary Patterns in Denmark

**DOI:** 10.3390/foods9091176

**Published:** 2020-08-25

**Authors:** Lisbeth Mogensen, John E. Hermansen, Ellen Trolle

**Affiliations:** 1Department of Agroecology, Faculty of Technical Sciences, Aarhus University, Blichers Allé 20, DK-8830 Tjele, Denmark; john.hermansen@agro.au.dk; 2National Food Institute, DTU, Technical University of Denmark, Kemitorvet, Building 201, DK-2800 Kgs. Lyngby, Denmark; eltr@food.dtu.dk

**Keywords:** diet, beef, environmental impact, greenhouse gases (GHG), land use, nutrition, life cycle assessment (LCA)

## Abstract

There is public focus on the environmental impact, and in particular, the emissions of greenhouse gases (GHG), related to our food consumption. The aim of the present study was to estimate the carbon footprint (CF), land use and nutritional impact of the different beef products ready to eat in different real-life dietary patterns. Beef products accounted for 513, 560, 409 and 1023 g CO_2_eq per day, respectively, in the four dietary patterns (Traditional, Fast-food, Green, and High-beef). The total CFs of these diets were 4.4, 4.2, 4.3 and 5.0 kg CO_2_eq per day (10 MJ)_,_ respectively. The Green diet had almost the same CF as the Traditional and the Fast-food diets despite having the lowest intake of beef as well as the lowest intake of red meat in total. A theoretical substitution of beef with other animal products or legumes in each of these three diets reduced the diets’ CF by 4–12% and land use by 5–14%. As regards nutrients, both positive and negative impacts of these substitutions were found but only a few of particular nutritional importance, indicating that replacing beef with a combination of other foods without a significant effect on the nutrient profile of the diet is a potential mitigation option.

## 1. Introduction

Red meat, including beef products, plays a central role in Western diets due to taste preferences, culinary tradition and social norms and has been recognised as an important provider of nutrients. Thus, the consumption of meat is typically high in most developed countries. However, it is well recognised that the production of meat and especially beef products is associated with a high load of greenhouse gas (GHG) emissions, particularly compared to plant-based foods [1,2,3]. In addition, beef products, like other types of meat, require a high land use [1,3]. The reason for the high GHG load of beef—also named a high carbon footprint (CF)—compared to other types of meat is mainly due to the digestion system of ruminants, which foster a high production of methane from the digestion of feed [4] but also a higher feed consumption per kg of the meat produced. There is an increasing demand for reducing the total climate impact of our diets as it has been estimated that today’s food supply is responsible for 26% of the anthropogenic GHG emission [3]. Therefore, there is an increasing focus on how dietary changes can reduce climate impact [5,6,7]. At the same time, there is growing evidence of an association between high consumption of red meat, especially of processed meat, and an increased risk of several major chronic diseases [8,9,10,11], and The World Cancer Research Fund International and the American Institute for Cancer Research recommend, no more than 350–500 g (cooked weight) of red meat, such as beef, pork and lamb, per week and only a small amount of, if any, processed meat [12]. However, like other foods, meat contributes important nutrients to the diet, and the impact of reducing beef in the diet needs to be evaluated both from an environmental and a nutritional point of view, taking into account the different eating habits.

Several studies have estimated the environmental impact of reduced meat or beef intake based on a comparison between a current and a theoretical diet [13,14,15]. Such work where the theoretical diet is composed with the aim to comply with the nutritional recommendations or with the aim to lower the total GHG emission often shows a considerable impact of reduced meat and, in particular, beef intake in the diet as summarised in recent reviews [16,17]. Differently, Vieux et al. [18] estimated GHG emission associated with self-selected diets in France and observed a less clear picture in that no major impact on GHG emission was observed when meat was substituted with fruits and vegetables on an iso-caloric basis. Likewise, Vieux et al. [19], analysing self-selected diets in five European countries, observed that the cluster with the lowest dietary GHG emission also had the lowest nutritional quality. These observations highlight the importance of considering real-life diets as a basis for evaluating the impact of diets on GHG emissions and nutritional quality simultaneously, as well as theoretical estimations of the impact of further changes.

Including GHG emissions related to the use and preparation on the household level has not been common practice in much work addressing the environmental impact. Thus, Hallström et al. [16] only identified two studies that included this. This was also the case in a recent study [20], whereas it was not included for meat in the recent work of Heller et al. [21]. In addition, the GHG emissions and land use also show a high variation depending on whether the beef originates from dairy or beef breed cattle [9,22], and little is known about the impact of the resource use and waste and losses in the food chain from the slaughterhouse until the food is ready for eating as well as about different types of food preparation on the total GHG emission of beef products compared to other products and the total diet

Thus, there is a need to better understand how the type of beef and the route from slaughterhouse to plate affects the GHG emissions and the land use demand of the diet considering different existing dietary patterns in real-life and to identify possible critical nutritional aspects when the intake of beef is reduced in different dietary patterns. The objective of the present work was therefore to estimate the GHG emission and land use of the beef products consumed in Denmark, also taking into account the resource use and losses in the chain from the slaughterhouse until the beef product is ready to eat, and to use these new data to investigate the carbon footprint and land use as well as the nutritional profile of the total diet of different Danish dietary patterns as well as the potential impact of replacing beef with other foods in these patterns.

## 2. Materials and Methods

The work takes the departure from the work of Mogensen et al. [22,23], describing the GHG emissions and land use for different types of beef leaving the slaughterhouse, and the work of Pedersen et al. [24], Knudsen et al. [25], and Trolle et al. [26], describing the average intake as well as dietary patterns in Denmark, based on data from the Danish National Survey on Dietary Habits and Physical Activity and thus factual intake reported by consumers.

### 2.1. Estimating Type of Beef Products Produced and Consumed in Denmark

A purchase study from 3000 Danish households [27] found that 60% of the beef bought in the Danish retail sector was produced in Denmark. The imported beef was mainly from neighbouring European countries and a very minor part (approximately 5%) from overseas countries. The beef originates from different types of cattle—young stock and cows from dairy breeds or specialised beef breeds—which have very different CF and land use [22,23] and which typically have particular use as regards the beef products consumed. The amount of bone-free meat produced in one year from different types of animals was estimated based on statistics of slaughtered animals in Denmark [28] combined with the corresponding part of bone-free meat for the different types of animals [22]. In total, nine types of cattle were included in the present work, covering dairy systems and beef breed systems as well as calves and older animals. Next, resource persons at the largest Danish beef slaughterhouse were interviewed by one of the authors regarding the estimated use of the different types of carcasses for the supply of minced meat, roasted meat, steaks and dice/strips, respectively. Based on this information we estimated how the different types of cattle contributed to the different types of beef products.

### 2.2. Environmental Impact of Different Beef Products

Previously, we have estimated the GHG emission and land use related to the beef originating from the aforementioned nine types of animals in Danish production systems [22,23]. This was carried out by a life cycle assessment (LCA) including the chain until the edible products leave the slaughterhouse and, therefore, includes both the emissions that occur at the farm and at the slaughterhouse and takes into account the use of by-products from the slaughtering process. All feed was considered imported to the herd and all manure was applied outside the herd. For each input like for example barley for feed, an independent LCA was used to estimate the environmental impact per kg barley. From the herd, there are emissions of methane (CH_4_) and nitrous oxide (N_2_O). Methane comes both from enteric fermentation and from manure handling. Nitrous oxide originates from the manure in the barn and the storage of manure. Finally, there is an indirect nitrous oxide emission through the evaporation of ammonia (NH_3_). The allocation of environmental impact between milk and beef in the dairy systems was based on a biophysical relationship (the theoretical feed requirement to produce milk and beef, respectively). For CH_4_ and N_2_O the equivalency factors to convert to CO_2_eq were 25 and 298, respectively. Further details on how GHG emission and land use of different beef products until slaughterhouse gate are estimated, are shown in Mogensen et al. [22,23].

The GHG emission and land use related to the nine types of beef animals in Danish production systems were used to estimate the environmental impact in terms of GHG emission and land use related to the primary production (ab slaughterhouse) irrespective of the countries from where the production took place since major importing countries were European with comparable GHG emission from dairy systems [29] as well as beef breed systems [30]. In the present study, we added impacts in the chain from the slaughterhouse until the beef was ready to eat by including GHG emission related to transport, packaging, storage, food waste, and cooking. Transport from shop to home was not taken into account in this study.

Transport of beef was estimated for transport in Denmark as regards the proportion produced in Denmark as well as accounting for international transport as regards the proportion produced outside Denmark. The beef meat produced in Denmark is transported from the slaughterhouse to the supermarket terminal and from there to the shops. Average distance from slaughterhouse to terminal was estimated to 220 km, and the average distance from the terminal to the shop was estimated to 81 km. International transport was based on the distance to countries from where import took place. The estimated distribution on import countries [31,32,33,34] is shown in Appendix A. Imported beef was mainly sourced from The Netherlands and Germany. Estimated GHG emissions related to transport [35,36] from different countries are shown in Appendix A.

The GHG emission of packaging was based on a Finnish study that assumed 0.10–0.20 kg CO_2_eq/kg of food from packaging production for all types of meat products [37]. Thus, an estimate of 0.15 kg CO_2_eq/kg beef from packaging production was used in this study. The GHG emission related to storage included both storage at the grocery and shop as well as at home (Appendix A). The energy use related to preparation and cooking of different types of beef products was estimated based on the assumptions [36,38,39] in Appendix A.

Land use for different types of cattle—young stock and cows from dairy breeds or specialised beef breeds—was based on Mogensen et al. [22,23]. For each type of cattle, a detailed feed ration was defined, and the related land use for production hereof could be estimated based on the average crop yield per feed crop. In the results section, we present land use both as total land, including permanent pastures and nature area for grazing and as use of arable land.

### 2.3. Environmental Impact of Different Foods on the Plate

#### 2.3.1. Farming and Processing

The GHG emissions from primary production on farms as well as the processing of each food item were based on existing literature values [40,41,42,43,44,45,46,47,48,49,50,51,52,53,54,55,56,57,58,59,60,61,62,63,64,65] (Appendix A). If possible, land use was from the same reference as GHG emission, e.g., for beef, pork, milk and milk products, bread and rice. For some food items, land use was based on Gerbens-Leenes and Nonhebel [66], e.g., eggs, beer, and wine. The average land use of fruit from Gerbens-Leenes and Nonhebel [66] was used for oranges, bananas, apples and pears, and their average land use for vegetables was used for beetroots, onions and lettuce. For a few food items (carrots, potatoes, peas and cabbages), land use was an estimate based on estimated crop yield.

#### 2.3.2. Transport and Storage

The GHG contribution from transport was estimated as described for beef products based on Appendix A. The GHG contribution from storage in the supermarket and at home was a rough estimate based on the assumed energy use, since detailed data for storage time and method were missing. These standard estimates were a GHG emission of 475 g CO_2_eq/kg food stored frozen, 100 g CO_2_eq/kg food stored cold and 60 g CO_2_eq/kg food stored dry. In Appendix A, the assumed type of storage for different food groups is provided as well as the GHG emission applied in this work.

#### 2.3.3. Cooking

Based on the expert knowledge and the applied cooking recipes from the Danish National Survey of Dietary Habits and Physical Activity (2005–2008) [24], an estimate of the proportion of each food which is cooked before eaten is given in Appendix A. The GHG contribution from the cooking of the different types of beef products is described in detail in Appendix A. This resulted in a weighted average for cooking of beef of 411 g CO_2_eq/kg beef. This number was used as an estimate for cooking other types of meat as well. Estimates of GHG emission for cooking, in general, were based on relatively few measurements of energy use for cooking [39]. Based on this, standard values for cooking were generated (Appendix A).

#### 2.3.4. Food Losses

Apart from the resource use during the supply chain for different products, the GHG emission and land use of the foods ready to eat are also impacted by the food waste in the chain and the cooking loss. In Appendix A, the level of avoidable food waste in the chain [67,68,69,70,71] is shown. Food waste during processing was assumed to be 2% based on literature values [72,73].

According to the literature, fresh vegetables, fruits and bread make up the largest quantities of food waste from the retail chain. In this study, a food waste of 6% was used for fresh vegetables, fruits, bread and cakes, 3% for dairy products, and 2% for other foods including eggs and meat. 

In households, there are both an avoidable food waste, i.e., discarding of edible foods, and an unavoidable food loss in the form of non-edible parts such as peels and bones, and a weight loss when preparing food at home. Knowledge about weight loss during cooking is needed in order to relate the amount of food eaten with the amount produced. Results for the unavoidable food loss [74,75] are shown in Appendix A and for the avoidable food loss in Appendix A. Estimates of food waste in households in Appendix A were based on a combination of the Danish inventories of waste disposal [71], an English estimate of the proportion of food waste (edible foods) in household waste in England and Wales [69,70] and a common European proposal for default values for this loss [67]. Appendix A shows for different food items the GHG emission and land use per food item ready to eat taking into account the whole chain from farm to fork including food losses along the chain.

### 2.4. Dietary Intake

The daily dietary intakes were based on data used by Knudsen et al. [25] for identifying Danish dietary patterns, derived from the Danish National Survey of Dietary Habits and Physical Activity (DANSDA), which is a cross-sectional survey of dietary habits and physical activity in a representative sample of the Danish population. Data in the present study cover the adult Danish population (18–75 years of age, *N* = 2025) collected in the years 2005–2008 [24]. Dietary intake was recorded for seven consecutive days in food diaries supplied with pre-coded answer options for the most commonly eaten foods, dishes and beverages in the Danish diet and open-answer categories for more seldom eaten foods or drinks. The food diary was organized according to the typical Danish meal pattern (breakfast, lunch, dinner and snacks). Portion sizes were estimated using household measurements or from photographs, as the amount eaten of the prepared foods and dishes. The diaries were scanned using The Eyes and Hands program (version 4.1, 1998; ReadSoft Ltd., Milton Keynes, Bucks, UK). Food and nutrient intakes were calculated, taking weight losses or gains and nutrient retentions through food preparation and cooking into account, using the software system GIES (version 0.995a, released 26 June 2005), developed at the National Food Institute, Technical University of Denmark, and the Danish Food Composition Databank [74]. For a more detailed description see Knudsen et al. [25].

Using these data, Knudsen et al. [25]—through principal component analyses—identified three different dietary patterns which we name Traditional diet, Green diet, and Fast-food diet, respectively. For the present study the average intake of approximately 25% of the adult population with the highest score on the Traditional, Green, and Fast-food pattern, respectively, was calculated. In addition to these patterns—for the purpose of the present work—a High-beef pattern was identified as the average intake among the 20% of the adult population with the highest intake of beef.

For comparison of the dietary patterns, the food and nutrient intake were adjusted to a daily intake of 10 MJ corresponding approximately to the average daily reference intake for men and women aging 31–60 years according to Nordic Nutrition Recommendations 2012 [76]. Based on the calculated average intake of foods and dishes consumed by the survey participants, the average intake of the specific foods and drinks was estimated for each dietary pattern (Traditional, Green, Fast-food, High-beef) and for the Average diet of the total adult population, reflecting the total actual combination of raw and cooked foods of their diets.

The composition of foods and food groups and the nutrient profile of the dietary patterns were compared to the Average diet, the Danish Food-Based Dietary Guidelines (FBDG) [77] and the recommended nutrient density (per 10 MJ) to be used for planning diets for groups of individuals at 6–65 years of age with a heterogeneous age and sex distribution from the Nordic Nutrition Recommendations 2012 [76]. Based on previous results of the Danish National Survey of Dietary Habits and Physical Activity [24,78], differences in the content of foods or food groups and nutrients between the dietary patterns and the Average diet by more than 10% were considered significant and are reported.

### 2.5. Effect of a Substitution of Beef Intake on the Environmental Impact and Nutritional Profile of the Diet

In order to estimate the potential climate impact of excluding beef from the diets, the effects on the nutritional profile, the GHG emission and land use of the Traditional, Green and Fast-food dietary patterns of replacing beef with pork, poultry, fish, eggs, cheese or legumes, respectively, were investigated. Beef was replaced with the same amount (weight) of pork, poultry, fish, eggs or cheese as beef present in each dietary pattern. The energy content of each dietary pattern was thereby in most cases kept approximately constant. The calculated energy intake was reduced by 1% by substitution with eggs in all dietary patterns, and by 1% by substitution with fish in the Fast-food pattern, while increasing by 1% by substitution with cheese in the Traditional and Fast food pattern. When replacing with legumes, to keep the total energy content constant at approximately. 10 MJ, the amount of beef was replaced by 1.84 g of cooked dried beans per g beef. Differences in the nutrient content of more than 5% between the original dietary patterns and the substituted patterns are reported.

### 2.6. Sensitivity Test

#### 2.6.1. Land Use Change

The food system is a major driver of changes in land use [79], and some studies include a GHG contribution from land-use change (LUC) in their calculation of GHG from the diet [16,80]. This can be performed both as a direct LUC (dLUC) and as indirect LUC (iLUC). In the European PEF guidelines [67] it has been suggested, that iLUC is not currently included in the climate calculations, as it is considered that the method and data for these calculations are too uncertain. As a sensitivity analysis, we have applied the method for including iLUC as suggested by Audsley et al. [50]. They found an average iLUC emission factor of 143 g CO_2_eq per m^2^ used for crop production. We used this emission factor for the arable land use per food item.

#### 2.6.2. Energy Based on Renewable Energy

The proportion of the electricity mix that is based on renewable energy is increasing year by year as this is a political goal. As a sensitivity test, we assumed that electricity used for cooking and storage was based 100% on renewables energy and investigated how this would affect the GHG emission of different food groups and the total diet.

## 3. Results

Table 1 shows the production of bone-free meat from different types of cattle in Denmark as well as the related GHG emission and land use per kg of beef supplied from the slaughterhouse. In addition, it is shown how the different types of slaughtered cattle contribute to the overall amount of different beef products produced and the resultant GHG emission and land use connected to each type of beef product. By far, the largest proportion of all beef originates from dairy systems (83%). Within dairy systems, dairy cows supply the largest share of minced meat and dice/strips (55% and 63%, respectively), whereas calves supply the largest share of steaks (59%).

Overall, the GHG emission of beef from dairy systems was estimated to 10.9 kg CO_2_eq per kg of beef without bones from the slaughterhouse against 23.3 kg CO_2_eq per kg of beef from beef breed systems. The total land use was also higher for beef from beef breed systems compared to dairy systems (52.6 and 14.1 m^2^ per kg of beef ab slaughterhouse, respectively). However, the requirement for arable land was not very different between beef from the two different systems: 16.1 and 14.1 m^2^ per kg of beef, respectively.

When considering the different beef products—minced, roasted, steak and dice/strips—taking the actual proportions of dairy and beef breed beef into account, relatively moderate differences were found ranging from 11.9 kg CO_2_eq per kg of minced meat to 15.3 kg CO_2_eq per kg of roasted beef at the point of leaving the slaughterhouse. The higher GHG emission of beef from roasted beef was due to a higher proportion of beef from the beef breed system.

Table 2 shows the GHG emission and land use of the different types of beef products ready to eat, accounting for resource use for transport and preparing as well as food waste, trimming and cooking losses in the process. Overall, the GHG emission of the beef products ready to eat was 56–66% higher than when leaving the slaughterhouse, highest for roasted due to higher trimming losses. Cooking and trimming losses were the most important contributions to the increase in the GHG emission per kg of beef ready to eat followed by food waste and energy for cooking. Contributions from packaging, transport and storage were minor contributors to GHG emissions after leaving the slaughterhouse.

The average intake of beef in the Average dietary pattern was 26 g/10 MJ (Table 3). The Traditional diet was very close to the Average diet, both in terms of total intake of beef and the distribution of beef products. In comparison, the Fast-food diet had a higher intake of beef, in particular of minced meat, whereas the Green diet showed lower intake of beef, in particular minced meat. The High-beef diet was characterised with the double intake of beef compared to the Traditional diet and in particular a higher intake of steak and minced meat. The resultant GHG emission from the intake of beef was almost directly related to the total intake since only minor differences existed in the GHG emission per kg of the different mixes of beef products. Thus, where the GHG contribution of beef in the Average diet was 529 g CO_2_eq per 10 MJ, it was approximately 10% higher for the Fast-food diet, 20% lower for the Green diet and almost double for the High-beef diet.

The dietary profile in the different dietary patterns is shown in Table 4. Compared to the Danish FBDG, which are health-based, the Danish average adult diet would improve by an increase of the content of fruit and vegetables, whole grain, fish and fats from vegetable sources (except coconut fat and palm oils) instead of animal source; and a decrease in red meat (beef, pork and lamb), alcohol-containing beverages and sugar containing food and beverages [77,81]. According to the Danish FBDG, potatoes are a positive part of a healthy diet, referring to boiled potatoes which are common in the Danish dietary culture—and not French fries. Rye bread and oatmeal are also common in the Danish diet and important contributors of whole grain, while wheat bread constitutes both white and whole-grain types of bread [77,81]. Compared to the Average diet, the Traditional diet included less wine and more rye bread and potatoes but also fewer vegetables and fruit more pork (and total red meat), butter (and butter-containing spread) and sugar (and sugar-like products). Other differences included less milk and wheat bread. The Fast-food diet included less beer and wine but also less rye bread and fewer potatoes, and less fish but, in particular, more soft drinks compared to the Average diet. In addition, the Fast-food diet included more wheat bread and less coffee. The Green diet compared to the Average diet is characterised by more vegetables, fruit, breakfast cereals and also fish, and less red meat, butter (and butter-like products), beer, wine and soft drink, but also more cake and fewer potatoes. Other differences are more tea and more milk and less coffee. The High-beef diet—apart from beef consumption—did not differ much from the Average diet, except for a higher content of potatoes but also a lower content of fruit (especially) and a higher content of beer, soft drink and slightly more wine and pork. Of all dietary patterns, the content of food and food groups of the Green diet is closest to the Danish FBDG.

Table 5 shows the GHG emissions and land use from the different dietary patterns, taking into account the whole chain from farm to fork including food losses along the chain. The Traditional, Fast-food and Green dietary patterns showed almost the same GHG emission, whereas the High-beef diet showed a higher emission. The Green diet and the Fast-food diet had a low GHG emission from beverages, and therefore the GHGs of the total Green diet and the total Fast-food diet were slightly lower than the Traditional and the Average diet. Additionally, it can be noted that the contribution from all food of the High-beef diet, compared to the Average, more or less corresponds to the difference in the contribution from beef. The GHG emission from the Green diet was 4% lower than the Average diet, and the land use requirement was 8% less. The High-beef diet had 16–19% more GHG emission and required 20% more land than the Green and Fast-food diets. Scaled to a yearly basis, the High-beef diet resulted in 190 kg higher CO_2_eq emission (12%) than the Average diet.

To produce the total average adult Danish diet of 10 MJ/day, there is a land use of 1580 m^2^ per person per year, which with the method described by Audsley et al. [50] provides a GHG contribution from iLUC of 226 kg CO_2_eq per person per year, corresponding to a 5% increase in the GHG of the diet. Whether the GHG contribution from iLUC was included or not in the present study did not affect the proportion of the GHG from different food groups, e.g., beef contributed 12% of the total diet GHG both when including iLUC or not.

In addition to the comparison of the climate impact of the diets an evaluation of the nutritional quality of the diets is highly relevant. Red and processed meat contribute with a high proportion of (≥30%) vitamins and minerals such as vitamin A, thiamine, niacin, B12, B6, zinc, selenium and protein of the average Danish diet. At the same time, red and processed meat, in particular, are major sources of saturated fat and sodium [82]. Table 6 shows the nutritional profile of the different dietary patterns. The nutrient content of the diets in National Dietary Habits and Physical Activity 2003–2008 was, in general, sufficient compared to the Nordic Nutrition Recommendations 2004 [24]. For most vitamins and minerals, the content was abundant (vitamin A, riboflavin, niacin and B12, calcium, phosphorus and iodine) or acceptable (vitamin E, thiamine, B6, folate and C and magnesium, zinc, selenium and potassium), low in vitamin D, however, and for some women of the fertile age also in iron [24]. Additionally, the contents of whole grain and dietary fibre were too low and saturated fatty acids were too high. This is reflected in the content of the Average diet in Table 6, and when comparing the nutrient content of the different dietary patterns the low and acceptable nutrient contents are of concern. Comparing the High-beef diet, which was also highest in total red meat, with the Green diet, showed that the High-beef diet did not improve the nutritional profile as regards the nutrients of concern. On the contrary, the Green diet was lower in saturated fatty acids and higher in dietary fibre and whole grain, vitamins E, C and D, folate, potassium and magnesium, and smaller improvements were seen for iron and selenium (9%), thiamine (7%) and vitamin B6 (6%). In addition, the content of protein, added sugar and zinc was about the same. Besides the lower intake of beef and total red meat, the Green diet was characterised by a higher content of oat flakes, rye bread, vegetables and fruit and of dairy products, fish and eggs, and a lower content of beer, wine, sweet beverages and coffee than the High-beef diet. These changes were apparently sufficient to account for the lower content of red meat as regards the critical nutrients. The nutrient profile of the Traditional and Fast-food diets was close to the Average diet. However, the contents of folate and vitamin C were lower in the Traditional diet, and the Fast-food diet was higher in added sugar, and lower in dietary fibre, whole grain, potassium, vitamin D and iron (close to 10%). Thus, the Green diet was closest to the recommended intake compared to all other diets; since saturated fatty acids, dietary fibre and whole-grain were improved, added sugar either improved or stayed at the same level, and the contents of most micronutrients were improved while others were approximately alike.

Any suggested dietary change in order to reduce the environmental impact of our diets should take into account the impact on the nutritive value of the diet. Due to the fact that the GHG emission per kg beef or per MJ beef is much higher than for most other regular foods, the potential impact of replacing beef with other foods in these patterns was investigated. Table 7 shows the impact on GHG emission, land use and content of the critical nutrients (which were identified in relation to nutritional evaluation of the different diets, Table 6) in relation to the three dietary patterns: Traditional, Green, and Fast-food, if the amount of beef in the diet was replaced with substitutes.

Using the substitutes in question here reduced the GHG emission by 4–12% and the land use by 5–14%. The largest impact on the GHG emission was obtained by using legumes or eggs as a substitute, whereas the largest impact on the land use, not surprisingly, was obtained by using fish—or eggs as the substitute. From a nutritional point of view, both positive and negative effects were observed, depending on the type of substitution. Focusing on the nutrients of concern, on the positive side, pork as a substitute increased thiamine in all dietary patterns (14–20%), eggs increased selenium in the Traditional and Fast-food diet (6–7%), fish increased vitamin D and selenium in all dietary patterns (35–66% and 9–15%, respectively), and legumes increased dietary fibre in all three dietary patterns (9–18%) and magnesium (5%) in the Traditional and Fast-food diet as well as folate and potassium (both 5%) in the Fast-food diet. On the negative side, all substitutes but pork reduced the content of zinc (7–10%) in all diets, except for cheese in the Traditional and Green diet. Poultry and fish also reduced the content of iron in the Fast-food diet (both 5%), and cheese also reduced the content of iron in both Traditional and Fast-food diets (6–7%). The content of vitamin D was reduced by legumes in the Traditional and Fast-food diets (6–7%) and by poultry and cheese in the Fast-food diet (both 5%).

## 4. Discussion

Of the different beef products—minced, roasted, steak and dice/strips—the roasted beef had the highest CF both as ready to eat and as slaughterhouse. The daily intake of beef was between 20 (the Green diet) and 50 g per 10 MJ (the High-beef diet) and accounted for 9 to 20% of the GHG emission of the different dietary patterns. The patterns also differed in the content of other food groups and in the nutrient profile, where the Green diet was closer to the FBDG and nutrient recommendations than the other diets which also differed in the content of especially fruit and vegetables and sweet beverages and alcohol-containing beverages. The total High-beef diet had the highest GHG emissions of all the dietary patterns while the Green and the Fast-food diets were only slightly lower than the Traditional diet and the Average diet. The substitution of beef with other protein-rich foods suggests this to be one possible way to obtain lower GHG emissions from the diet.

### 4.1. Importance of Including Food Preparation and Food Waste and Losses in the CF of the Diets

In the present study, GHG emissions caused by energy for cooking were included. For beef, cooking contributed 2.7% of the total CF per kg of beef ready to eat. In total, 5% of the GHG emissions of the diet were caused by cooking. In the Average diet, the highest contribution to cooking (36%) came from the cooking of coffee and tea, another 25% came from the cooking of meat and fish, and 23% from the cooking of vegetables.

If all energy used for both cooking and storage was 100% based on renewable energy, the GHG emissions of the diet could be reduced by 12% and the GHG emissions from food by 10%. However, this would only lead to small differences in the relative impact of different food groups in that the proportion of GHG emissions due to consumption of vegetables would be reduced from 5 to 4% of the total GHG emissions, whereas the proportion of GHG from beef would increase from 12 to 13% of the total GHG emissions from the diet.

In the present study, 12–13% of the GHG emissions and 13% of the land use of the diets were caused by avoidable food waste. For beef, ready to eat, 12% of the GHG emissions and land use were caused by avoidable food waste. Considered per kg of food item, avoidable food waste differs significantly between food groups: from 21 to 24% of GHG emission for fruit, bread and vegetables ready to eat to 14% of GHG emission for meat and 6–8% of GHG emission for sweets and milk ready to eat. Due to this difference, the contribution per food group to the total GHG emissions of the diet will differ, whether GHG emissions from food waste are included or not. However, the differences are relatively small, and meat contributed to 31% of the GHG emissions from the diet whether food waste was included or not. Overall, the reduction of avoidable food waste can be seen as a significant mitigation option for reducing GHG emissions related to the diet as also pointed out by the EAT-Lancet Commission [7] and other studies, e.g., [79], with a potential of a reduction of 12–13% in the Danish diets. 

While there are major differences in the CF, whether considered per kg food in the supermarket or per kg food ready to eat, the overall ranking of GHG emissions of different food was not changed as illustrated in Figure 1. However, as the reduction in both energy for cooking and food losses are potential mitigations, it is important to include these contributions when studying GHG emissions of the diet.

### 4.2. The Nutritional Evaluation

Dietary changes in accordance with FBDG are promoted in countries all over the world in order to improve health and reduce the risk of chronic diseases [83]. The increasing focus on dietary changes based on the increasing demand for reducing the climate impact of the diet provides the opportunity of dietary changes that at the same time might fulfil the FBDG and improve the nutritional quality and health effect of the diet.

In the present study, we estimated the potential reduction of the CF of the diet by rather simple substitutions where we calculate the effect of replacing beef with one other food at the time. These estimations indicate that a reduction of beef intake in real-life dietary patterns can maintain the nutritional quality of the diets if the beef is exchanged with a combination of other protein-containing foods. However, the results also indicate that depending on the amount of beef and the percentages of reduction, the content of zinc and to some extent iron will need special attention, especially because substitution with pork was less relevant, since the content of pork before substitution, and thereby of red meat, was above or around the recommended maximum intake of 500 g per week, corresponding to on average 71 g per day according to the Danish FBDG [77].

The recent update from the World Cancer Research Fund International and the American Institute for Cancer Research (2018) recommends, based on evidence for the risk of red meat causing different kinds of cancer, no more than 350–500 g (cooked weight) of red meat, such as beef, pork and lamb per week (on average 50–71 g per day) and only a small amount of, if any, processed meat [12]. Other studies including systematic reviews indicate positive associations between intake of red meat and processed meat and the risk of stroke, cardiovascular mortality and all-cause mortality [8,9,10,84] myocardial infarction [84] and diabetes type 2 [8,11,84]. However, the magnitude of association and quality of evidence has been debated as described by Lassen et al. [85] and no overall assessment is available of the total evidence commissioned by national food or health authorities or international food and health organisations, using methods as used by The World Cancer Research Fund International and the American Institute for Cancer Research.

Cooked dry brown beans were used for the substitution of beef meat with legumes. Since the nutrient content of different types of legumes varies [86], different combinations might result in other nutrient profiles. Replacing 50% of total meat in the Swedish average diet with grain legumes maintained the energy content and most nutrient contents within the Nordic Nutrition Recommendations while the dietary fibre and folate intake (which is currently low in the Swedish diet) increased [86]. More types of legumes have entered the market in recent years, and with more research on nutrient content, it might be possible to additionally optimise the substitution of larger meat contents than investigated in the present study. The Swedish study suggests focusing on including foods high in iron when replacing meat in diets for women of the childbearing age and pregnant women while the present study also suggests focusing on including a combination of other foods high in zinc. The substitution with legumes in the present study improved the content of saturated fat and dietary fibre besides several micronutrients.

The nutritional quality of the four dietary patterns was evaluated both in relation to the content of foods and food groups related to the Danish FBDG and with regards to the nutrient content. The distribution of nutrient intakes of the average diet of adults was used to point out nutrients of concern for inadequacy. As expected, the Green diet, which was closer to the FBDG than the other diets, was also closer to the nutrient recommendations.

The selenium content of the different diets was almost similar and evaluated against the recommended intake from Nordic Nutrition Recommendations 2004 (NNR2004 [87], the content of the average diet was acceptable. Although the recommended intake of selenium had increased from 40 to 57 µg per 10 MJ in Nordic Nutrition Recommendations 2012 (NNR2012) [76], the selenium content is most likely acceptable since the selenium data in the food composition databank have also been updated [78]. In addition, substitution with fish improved the content of selenium in the diets by 9–15% and with eggs by 6–7% in the Traditional and the Fast-food diet. 

Low intake of vitamin D is a well-known feature of Nordic diets [88] and is therefore expected despite the dietary pattern. Only a substitution with fish increased the content. Fortification and supplement use are other strategies to be used to increase the intake, since the effect of sunlight on the human skin can only provide production of vitamin D in the summer season in Denmark, as in other Nordic countries [88]. These findings are supported by the calculated nutrient content of the national adaption of the EAT-Lancet reference diet [85]. This Danish-adapted healthy plant-based diet based on the EAT-Lancet reference diet has a low content of meat, especially red and processed meat, sugar-containing foods and beverages, salt and alcohol, a moderate content of seafood, milk, poultry, eggs and vegetable oils, and a high content of vegetables (incl. dark green and red-orange vegetables) fruit, cereals and bread, legumes and nuts and seeds to fulfil the nutrient recommendations for planning diets per 10 MJ.

It is an advantage of the present study that it is based on the actual dietary intakes of a national representative dietary survey. However, it is well-known that dietary surveys have their limitations. Uncertainties in the calculated intake are due to uncertainties in portion size estimation and in self-reported choice of food when recording the dietary intake, which often results in over- and under-reporting. Under- and over-reporting within the dietary survey was calculated using the Goldberg equation (adopted by Black) to 17.7% and 1.4%, respectively [89,90,91]. Using the average intake adjusted to 10 MJ reduces the effect of under-reporting in the comparison of diets, although only to a certain degree since under-reporting might be more pronounced for some food groups than others [92,93]. However, the excessive energy intake which is plausible for a considerable part of the Danish adult population, since the prevalence of overweight is around 50% or more [94] is not reflected when the diets are adjusted to 10 MJ.

### 4.3. The “Un-Expectedly” High GHG Emissions from Food in the Green Diet

The Green diet was closest to the nutrient recommendations and the Danish FBDG. Compared to the Average diet, the Green diet was more plant-based, with 48% more vegetables, 60% more fruit and 53% more morning cereals. The Green diet had a slightly higher GHG contribution from the foods than the Average diet, despite having the lowest intake of beef as well as of red meat in total. Compared with the Average diet, the Green diet daily saved 270 g of CO_2_eq from meat and 50 g of CO_2_eq from eggs and fat. However, this was counterbalanced by more CO_2_eq from fruit, vegetables, bread, milk and fish. Similar results were found by Vieux et al. [95], studying self-selected diets of French adults, showing that plant-based diets of high nutritional quality were not lower in GHG emissions. This is in line with results from a simple modelling study [96] of the average adult diet to fulfil the Danish Food-Based Dietary Guidelines and Nordic Nutrition Recommendation as well as with more comprehensive modelling of the Traditional, Fast-food and Green dietary patterns to fulfil the Danish Food-Based Dietary Guidelines and Nordic Nutrition Recommendation in a previous study [26]. These changes in the diets where the red meat was reduced but total meat content was approximately unchanged resulted in a small GHG reduction of 2–4% for the three diets. However, if at the same time the most climate-friendly food items within the food groups were chosen, the GHG reduction became 22–29% for the three diets. A study modelling the UK adult diet to conform to WHO recommendations found a 17% reduction in GHG emissions [97]. A systematic review by Aleksandrowicz et al. [17] found a median reduction of 12% of healthy diets, which follows dietary guidelines, while Hallström et al. [16], in their systematic review based on almost the same studies, found a reduction by 0–35% for healthy diets compared to current diets. Additionally, Ritchie et al. [98] demonstrated that following the majority of current national diet guidelines is inconsistent with the climate target. Therefore, in order to take the environment and climate into account, future dietary guidelines should be reframed. A few countries have undertaken this recently. In 2019, FAO/WHO developed The Guiding Principles for Sustainable Healthy Diets which “are food based, and take into account nutrient recommendations while considering environmental, social/cultural and economic sustainability”, and they encouraged countries to start the process of implementation of sustainable healthy guidelines [99].

### 4.4. Methodology for Estimating GHG Emissions of Real-Life Diets

In the present as well as in similar studies (e.g., Hartikainen and Pulkkinen [37] and Macdiarmid et al. [100]), the total GHG emissions of the diet are the sum of the product of GHG emission per food item and daily intake per food item where the GHG emission by each food item is based on individual LCA studies. Although, the most suitable references are used, there will always be some uncertainty in the GHG emission value applied per food item due to differences in the methods applied, difference in the year of data for primary production etc. Especially when it comes to the more processed foods like some sweets and beverages, the number of LCA studies behind these food groups is much lower and the uncertainty for these GHG estimates per item similarly higher. Furthermore, assumptions often have to be made about food items for which no LCA values are available, but which are assumed to be of similar magnitude as food items for which LCA values are published.

Nevertheless, overall different studies arrive at almost the same aggregated values for the entire diet. In the present work, the GHG emissions from the diets varied from 1510 kg CO_2_eq per person per year (the Fast-Food diet) to 1830 kg CO_2_eq per person per year (the High-Beef diet). This range is similar to the results from other recently published work as regards Danish diets, (1590 kg CO_2_eq according to Bruno et al. [20]), and Swedish diets (2 t CO_2_eq of which 11% were emissions from tropical deforestation according to Cederberg et al. [80], and 2.2 t CO_2_eq from Moberg et al. [101]). The estimates by Bruno et al. [20] are based on supply data and a lower energy level (2000 kcal) of the diet than the present study and, like the present study, do not include GHG emissions related to land use changes. Cederberg et al. [80] and Moberg et al. [67] on the other hand included LUC carbon emissions in their work. If iLUC is included, the yearly emission per person (10 MJ) increased in our work by from 214 kg CO_2_eq (Green diet and Fast-food diet) to 254 kg CO_2_eq (High-beef diet), which is in line with the results from Cederberg et al. [80] (220 kg CO_2_eq from dLUC as regards Swedish diets) although estimated with a different method.

### 4.5. The Importance of Beef in the Diet and Mitigation Options

Recognizing that other issues as liking, pleasure, convenience and price are strong motivators of dietary choices [102], the present study is limited to investigate the potential impact on CF and nutritional value of the diets in order to be able to suggest changes that take both the need of CF reduction and nutritional improvements into account. How to promote such changes effectively is challenging and not within the scope of this paper.

The CF of the Green diet or the Fast-food diet was around 16–19% lower than the CF of the High-beef diet. In the present study, beef products were responsible for from 9% (Green diet) to 20% (High beef diet) of the GHG emission of the total diet. Substitution of beef with the same weight of other animal-based food (pork, poultry, eggs, cheese) or substitution with the same amount of energy from legumes reduced the GHG emission by 4–12% and the land use by 5–14%. Other studies found a higher effect of replacing ruminant-based meat with meat from monogastrics. In a review, Aleksandrowicz et al. [17] found a 21% reduction (median) by this substitution, and Hallström et al. [16] found that substitution of meat from ruminants by meat from poultry and pork could reduce the GHG emission by 20–35%. The lower effect in the present study could be explained by the relatively low CF applied for beef consumed in the Danish diets (12–15 kg CO_2_eq/kg beef ab slaughterhouse and 19–25 kg CO_2_eq/kg beef ready to eat for the different types of beef products), because a high proportion of the beef in the Danish diets originates from dairy cattle and that almost all beef comes from highly intensive Western European production systems. In addition, we saw a relatively low intake of beef of 20–28 g of the Green, Average, Traditional and Fast-food diets, and 50 g of the Danish High-beef diet, where the High-beef intake is comparable to the average intake of US self-reported diets [103].

Since the Danish diets have a much larger proportion of meat from pork than beef, a larger reduction of GHG emissions of the diet might be seen if the total meat intake of the Danish diet was considered. For further GHG reduction, a reduction of all animal based food could be considered as exclusion of animal-based products from the current diets has been found to have a huge potential to reduce land use by up to 76% and GHG emissions by up to 55% [3,5,16,17,103]. However, both the mitigation effect and the health effect of replacing animal-based food with plant-based food will depend on the actual substitutions and the composition of foods within the total changed diet.

In the present study, sugar, sweets and beverages (excluding milk and juice) contribute with 27% of the GHG emissions of the Average diet. The relatively high intake per day gives this substantial contribution to the GHG emission of the diet. Saxe et al. [104] and Moberg et al. [101] found similar GHG emissions caused by the consumption of sweets, snacks and drinks (excluding milk), and also Kanemoto et al. [105] showed that consumption of confectionary and alcohol contributes significantly to climate change. At the same time, a high intake of soft drinks, sweets and alcohol is negatively associated to health [106,107,108,109] and both NNR2012 [76] and WHO recommend to limit the intake hereof [110]. These foods are often related to excessive energy intake and a reduction of CF of the diet by up to 10% has been estimated if energy intake was balanced to energy expenditure [16]. To keep the energy level of the dietary patterns at 10 MJ substitution of sweets and beverages with other foods as whole grain, fruits and vegetables is needed and will affect the reduction of the CF of the diet.

## 5. Conclusions

The CF of different beef products—minced, roasted, steak and dice/strips—ready to eat ranked from 18.7 kg CO_2_eq per kg of minced meat to 25.5 kg CO_2_eq per kg of roasted beef. The differences were due to different sourcing of beef from dairy and beef breed-based systems and the fact that trimming and cooking losses were higher for roasted beef than for other types. Overall food waste accounted for 12% of the CF of beef, ready to eat. In four Danish dietary patterns named Traditional, Fast-food, Green and High-beef, the daily intake of beef was 25, 28, 20 and 50 g per 10 MJ, respectively. Beef products accounted for 12, 14, 9 and 20% of the GHG emission of the diets, respectively. Whether iLUC was included or not in the present study did not affect the proportion of GHG emissions from different food groups, e.g., on average, beef contributed with 12% of the GHG emissions of the total Average diet, both when including iLUC and when not. Likewise, the ranking of CF of different food groups per kg food in the supermarket and per kg food ready to eat was not dependent on whether a contribution from food preparation and food losses was included in estimating the GHG emissions or not. The nutritional quality of the different self-selected dietary patterns varies, with the Green diet being the healthiest by being closer to fulfilling the FBDG and nutrient recommendations. The total High-beef diet had the highest GHG emissions of all the dietary patterns, 16–19% higher than the Green and the Fast-food diets, which were only slightly lower than the Traditional diet and the Average diet, since a higher intake of other foods than beef contributed to the total GHG emissions from the diets. In addition, the substitution of beef with other protein-rich foods showed this to be one possible way to obtain lower GHG emissions from the three identified Danish dietary patterns (Traditional, Fast-food and Green) by up to 12%, highest for substitution with egg and legumes. At the same time, it seems possible to keep or even improve the nutritional quality of the diets. However, additional dietary changes in the direction of a more plant-based diet are needed for a larger climate impact, but such changes in dietary habits are even more challenging than beef reduction, and the changes need additional attention to be sure that the diets are healthy and sufficient.

## Figures and Tables

**Figure 1 foods-09-01176-f001:**
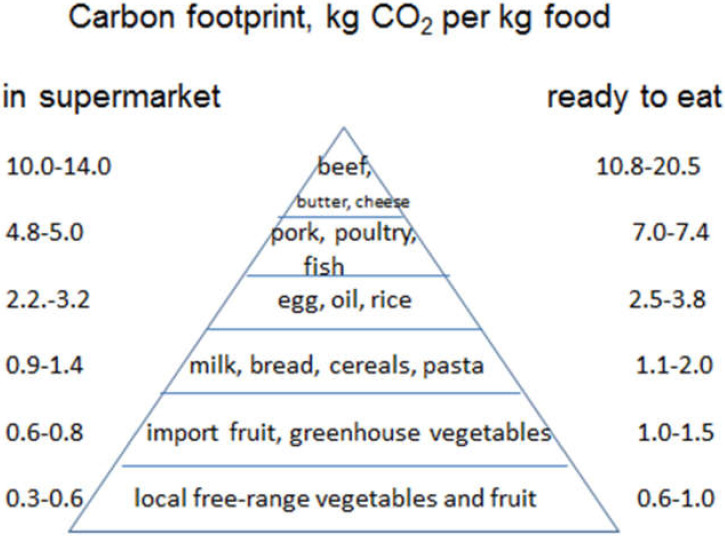
Carbon footprint (kg CO_2_eq) of different food groups per kg food in the supermarket and per kg food ready to eat,—including the contribution from farm to fork, food preparation and related losses.

**Table 1 foods-09-01176-t001:** Origin of different types of Danish-produced beef products, the related greenhouse gas (GHG) emission and land use estimated at the point of leaving the slaughterhouse and the contribution to different types of beef products.

Total Production, Ton Beef without Bones		Ab Slaughterhouse, per Kg Beef ^(5)^			Contribution to Different Types of Beef Products, % Distribution			
		GHG, kg CO_2_eq	Total Land Use, m^2^	Arable Land Use, m^2^	Minced	Roasted	Steak	Dice/Strips
Dairy based ^(1)^								
Calf	22,171	10.4	14.1	14.1	9.4	26.7	59.1	13.4
Young bull	17,529	10.5	15.5	15.5	22.4	16.9	18.7	0
Dairy cow	34,500	11.1	12.7	12.7	55.0	16.7	0	62.5
Other ^(2)^	4081	-	-	-	6.1	2.5	1.2	5.5
Total dairy	78,281	10.9	14.1	14.1	92.9	62.7	79.0	81.4
Intensive beef breed ^(2,3)^								
Calf	2946	32.0	57.6	24.4	1.3	7.1	3.1	3.6
Young bull	3196	31.0	55.1	25.3	1.3	7.6	3.4	3.8
Heifer	1680	30.8	57.1	21.5	0.7	4.1	1.8	2.0
Cow	6596	11.3	21.7	7.5	2.8	15.9	7.0	8.0
Extensive beef breed ^(4)^								
Young bull	1023	41.9	168.6	18.3	0.4	1.2	2.7	0.6
Heifer	245	45.8	240.5	19.0	0.1	0.3	0.7	0.1
Cow	869	12.9	62.9	5.1	0.4	1.0	2.3	0.5
Total beef breed	16,554	23.3	52.6	16.1	7.0	37.2	21.0	18.6
Total	94,835	13.1	20.8	14.5	100.0	100.0	100.0	100.0
GHG ab slaughterhouse, kg CO_2_eq/kg					11.9	15.3	13.5	13.1
Total land use, m^2^/kg					15.3	23.8	18.8	18.1
Arable land use, m^2^/kg					13.8	15.0	14.8	13.6

^(1)^, When allocating the environmental impact of the dairy system, milk is the main product that pays the major part of the environmental cost of the system. The co-products, meat from the dairy cow sent to slaughter and the newborn calf, are only responsible for the marginal cost of their production. Likewise, in beef breed production represented by Limousine (Intensive) and Scottish Highland (extensive), the weaned calves raised for slaughter are the main product that is responsible for the major part of the environmental cost of the system, whereas the co-product, meat from the cow sent to slaughter, only pays the marginal cost for its production (After Mogensen et al. [22]); ^(2)^, Mainly organic cows and steers; ^(3)^, Intensive beef breed production represented by Limousine; ^(4)^, Extensive beef breed production represented by Scottish Highland; ^(5)^ After Mogensen et al. [22,23].

**Table 2 foods-09-01176-t002:** Greenhouse gas (GHG) emissions and land use of beef products on the plate (per kg of beef ready to eat) depending on the type of beef, and contributiors from the food chain.

GHG Emission and Land Use, per Kg Beef Ready to Eat and the Contributors within the Food Chain	Minced	Roasted	Steak	Dice/Strips
**GHG, kg CO_2_eq, total**	18.7	25.5	21.1	20.6
*GHG, kg CO_2_eq, contribution from:*				
Meat ab slaughterhouse	11.9	15.3	13.5	13.1
Packaging and storage	0.3	0.3	0.3	0.3
Transport ^(1)^	0.1	0.2	0.2	0.1
Energy for cooking	0.4	0.5	0.3	0.4
Food waste ^(2)^	2.2	2.8	2.4	2.4
Trimming and cooking loss ^(3)^	3.7	6.5	4.4	4.3
**Land use, m^2^**				
Total land	22.4	37.2	27.7	26.7
Arable land	20.1	23.5	21.8	20.1

^(1)^, National, as well as international transport, included and reflecting that roast and steak generally are produced to a higher degree from imported beef than are minced and dice/strips; ^(2)^, From food losses in the chain—2, 2 and 11% in processing, retail and household, respectively; ^(3)^, Includes losses by trimming depending on the type of beef: 0, 8.5, 1 and 1% for minced, roasted, steak and dice/strips, respectively, and finally 20% weight loss at cooking for all types of beef.

**Table 3 foods-09-01176-t003:** Consumption of different beef products in the Average diet and in different dietary patterns and the related greenhouse gas (GHG) emission and land use from beef scaled to a total daily intake of 10 MJ per adult person, per person per day including the whole chain.

Dietary Patterns	Average Diet	Traditional	Fast-Food	Green	High-Beef
**Beef intake, total cooked, g**	26	25	28	20	50
*Hereof intake from*					
minced, g	14	13	17	11	24
roasted, g	3	3	3	2	6
steak, g	8	8	7	6	19
dice/strips	1	1	1	1	1
**Emissions from beef intake**					
GHG emission, CO_2_eq, g	529	513	560	409	1023
(GHG, kg CO_2_eq/kg beef mix)	(20.4)	(20.5)	(20.0)	(20.5)	(20.5)
Land use, from beef intake, m^2^	0.68	0.66	0.71	0.53	1.32
Land use, from beef intake, m^2^ arable	0.55	0.53	0.58	0.42	1.06
(m^2^ total/kg beef mix)	(26.1)	(26.4)	(25.2)	(26.2)	(26.3)
(m^2^ arable/kg beef mix)	(21.1)	(21.2)	(20.8)	(21.2)	(21.2)

**Table 4 foods-09-01176-t004:** Average food content (g) of different dietary patterns for adult persons scaled to an energy intake per day of 10 MJ, g/day and Danish Food-Based Dietary Guidelines (FBDG) [77].

Dietary Pattern Food Item (g per 10 MJ)	FBDG	Average	Traditional	Fast-Food	Green	High-Beef
**1. Milk products (incl. cream)**	250–500	343	301	362	381	335
**2. Cheese**	15–25 *	35	35	34	36	33
**3. Cereals, bread and dry pasta**						
Oat flakes + morning cereals		15	13	16	23	13
Wheat bread		105	96	127	98	106
Rye bread		71	82	47	76	66
Rice, dry		17	17	19	20	18
Durum pasta, dry		18	19	27	18	19
**4. Potatoes**		88	100	58	70	103
**5.Vegetables, total**	>300	171	138	164	253	158
**6. Fruit, total**	>300	214	150	172	343	170
**7. Juice**		80	58	95	92	72
**8. Red meat, total #**	<71 **	103	117	100	77	133
Beef		26	25	28	20	50
Pork		75	90	70	55	80
**9. Fish**	50 ***	22	23	15	31	18
**10. Poultry**		22	20	25	23	19
**11. Eggs**		13	12	9	14	11
**12. Fats**						
Butter (+ butter-containing fats)		14	21	13	10	17
Vegetable oils		21	23	20	19	20
**13. Sugar and sweets**						
Sugar and sugar-like products		14	20	11	15	14
Sweet products		78	81	87	87	78
Ice cream, mousses		13	14	15	14	15
Candy and chocolate		22	20	33	24	23
Cakes		43	46	39	49	40
**14. Beverages**						
Beer		187	200	130	79	245
Wine		102	75	38	92	111
Alcohol, e.g., whiskey		5	5	3	4	5
Lemonade, (ready to drink)		90	102	124	75	97
Soft drinks like cola		173	150	344	126	198
Water (tap)		857	549	743	1242	727
Bottled water		128	90	120	142	120
Coffee (ready to drink)		683	708	315	564	716
Tea (ready to drink)		163	105	148	296	110

*, Depending on the fat content; #, incl. lamb; **, 500 g red meat per week; ***, 350 g per week (200 g oily fish).

**Table 5 foods-09-01176-t005:** Greenhouse gas emission (GHG) and arable land use (LU) of the different dietary patterns and the content of food and drinks, per person per day (10 MJ) (including the contribution from food waste).

Dietary Patterns	Average	Traditional	Fast-Food	Green	High-Beef
**GHG, kg CO_2_eq/day (10 MJ)**					
Total diet	4.5	4.4	4.2	4.3	5.0
Total food	3.5	3.5	3.5	3.6	4.0
Contribution from					
Milk and cheese	0.8	0.8	0.8	0.9	0.8
Bread and cereals	0.3	0.3	0.3	0.3	0.3
Vegetables and potatoes	0.2	0.2	0.2	0.3	0.2
Fruit and juice	0.3	0.2	0.3	0.5	0.3
Meat	1.26	1.32	1.28	0.99	1.77
Fish	0.16	0.16	0.11	0.22	0.14
Eggs and fats	0.3	0.3	0.2	0.2	0.3
Sugar and sweets	0.2	0.2	0.3	0.3	0.2
Total beverages *	1.0	0.9	0.7	0.7	1.1
Contribution beef	0.5	0.5	0.6	0.4	1.0
(contribution beef, % of diet)	12	12	14	9	20
**Land use, m^2^/day (10 MJ)**					
Total diet	4.3	4.4	4.1	4.1	4.9
Total food	3.8	3.9	3.8	3.7	4.3
Contribution from					
Milk and cheese	0.8	0.7	0.8	0.8	0.7
Bread and cereals	0.4	0.4	0.4	0.4	0.4
Vegetables and potatoes	0.1	0.1	0.1	0.2	0.1
Fruit and juice	0.2	0.2	0.2	0.4	0.2
Meat	1.6	1.7	1.6	1.3	2.2
Fish	0	0	0	0	0
Eggs and fat	0.4	0.5	0.3	0.3	0.4
Sugar and sweets	0.3	0.3	0.4	0.4	0.3
Total beverages *	0.6	0.5	0.3	0.5	0.6
Contribution beef	0.6	0.5	0.6	0.4	1.1
(contribution, % of diet)	13	12	14	10	22
**Per year**					
GHG, tons CO_2_eq/year	1.64	1.61	1.51	1.58	1.83
(relative)	100	98	93	96	112
LU, 1000 m^2^/year	1.58	1.61	1.50	1.50	1.78
(relative)	100	102	95	95	112

*, Total beverages include: beer, wine, alcohol, lemonade, soft drinks, tap water, bottled water, coffee and tea.

**Table 6 foods-09-01176-t006:** Nutritional profile in different dietary patterns and indications of nutrients/diet-characteristics that are to be improved (bold) or at least maintained by a change of the diet (italic).

Dietary Patterns	Average	Traditional	Fast-Food	Green	High-Beef	Reference Values **
Intake beef, g/10 MJ	26	24	29	20	50	
**Nutrients (and whole grain)**	**Macronutrients and whole grain**					
Protein, total g/10 MJ	84	82	82	85	86	
Protein, E%	15	14	14	15	16	
Fat, total, E%	35	38	35	33	37	25–40 (32–33)
**Saturated FA *,** E%	**15**	**16**	**15**	**13**	**16**	<10
Mono-unsaturated FA *, E%	12	13	13	12	13	10–20
Polyunsaturated FA *, E%	5	5	5	5	5	5–10
*Added sugar*, E%	10	10	12	9	9	<10
**Dietary fibre,** g/10 MJ	**23**	**22**	**20**	29	**21**	25–35
**Whole grain,** g/10 MJ	**55**	**56**	**47**	**67**	**49**	75 ***
**Per 10 MJ**	**Vitamins and minerals**					
Vitamin A, RE	1255	1324	1075	1425	1200	800
Riboflavin, mg	1.8	1.7	1.7	1.9	1.7	1.4
Niacin, NE	33	33	30	33	35	16
Vitamin B12, µg	5.7	5.9	4.9	6	5.8	2
Calcium, mg	1180	1053	1215	1325	1130	1000
Phosphor, mg	1555	1497	1530	1640	1545	800
Copper, mg	4.3	3.3	4	5.6	3.7	1
Iodine, µg	215	198	207	231	201	170
*Vitamin E*, a-TE	**8**	**8**	**8**	10	**7**	9
*Thiamine*, mg	1.4	1.4	1.3	1.4	1.3	1.2
*Vitamin B6*, mg	1.6	1.5	1.5	1.7	1.6	1.3
*Folate*, µg	348	309	323	424	323	300–450
*Vitamin C*, mg	122	101	114	160	109	80
*Potassium*, g	3.7	3.5	**3.3**	4.1	3.6	3.5
*Magnesium*, mg	387	363	350	422	374	320
*Zinc*, mg	12	11	11	12	12	9–11 #
*Selenium*, µg	47	46	44	50	46	57
**Vitamin D,** µg	**3.7**	**3.7**	**2.8**	**4.2**	**3.4**	10 #
**Iron,** mg	**11**	**11**	**10**	**12**	**11**	11–16 #

E%, percentage of energy (excl. energy from alcohol); * FA, Fatty Acids; **, Reference values per 10 MJ from Nordic Nutrition Recommendations (NNR) 2012 [76], micronutrients: for use in dietary planning and thus typically higher values than the actual demand in individuals; ***, Danish recommendation [77,81]; #, The lower range is a typical Recommended Intake (RI) for adults.

**Table 7 foods-09-01176-t007:** Changes in greenhouse gas (GHG) emission and land use and an indication of relevant nutrients/food characteristics that change at least 5% by substituting beef products with other foods in the dietary patterns (Traditional, Green, and Fast-food) adjusted to the same energy intake (10 MJ/day). Positive changes with regard to nutrients of concern are indicated in bold, negative changes with regard to nutrients of concern are indicated in **bold** and *italic*.

Substitute Foods	Relative GHG Emission	Relative Land Use (Arable)	Increase in Nutrient Content	Reduction in Nutrient Content
Pork	0.91–0.94	0.93–0.95	**Thiamine** in all patterns	B12 in Fast-food.
Poultry	0.91–0.94	0.92–0.94	Polyunsaturated fat in Fast-food	***Zinc*** in all patterns, B12, ***vitamin D*** and ***iron*** in Fast-food
Eggs	0.88–0.92	0.88–0.91	**Selenium** and **vitamin D** in Traditional and Fast-food	***Zinc*** and niacin in all patterns, ***B6*** in Traditional and Fast-food
Fish	0.91–0.94	0.86–0.90	**Vitamin D**, B12, iodine and **selenium** in all patterns; **vitamin E** in Traditional and Fast-food.	***Zinc*** in all patterns and ***iron*** in Fast-food
Cheese	0.94–0.96	0.93–0.95	Calcium in all patterns; Phosphorous and ***saturated fat*** in Traditional and Fast-food; vitamin A in Fast-food	***Iron*** in Traditional og Fast-food; ***zinc*** and ***vitamin D*** in Fast-food.
Legume	0.88–0.92	0.93–0.95	**Dietary fibre** in all patterns; **magnesium** in Traditional and Fast-food; **folate** and **potassium** in Fast-food	***Zinc***, B12, monounsaturated fat, niacin, and ***vitamin D*** in all patterns. **Saturated fat** in Fast-food

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
