# Peer review of "The Climate and Nutritional Impact of Beef in Different Dietary Patterns in Denmark"

_foods, 2020, doi:10.3390/foods9091176_

Round 1
Reviewer 1 Report
This is a curious study showing the impact of beef in different dietary patterns on the emissions of greenhouse gases. I have some minor comments:
The sentence in lines 21 and 22 is non informative. It must be more specific or be deleted.
The Danish population is based on a relatively old study from 2005-2008, more than 10 years ago. More recent data would be more useful; for instance, EFSA has probably recent studies available.
Table 1: why such large difference in GHG between Heifer and Cow?
Author Response
Dear Editor and reviewer 1,
Thank you very much for the helpful comments, suggestions and questions from you.
We have now revised the paper according to these and have been able to change almost everything in the suggested way as can be seen in the List of changes below.
Looking forward to hear further decisions about this paper.
Yours sincerely,
Lisbeth Mogensen
Authors reply to the Review Report
- Reviewer 1 (R1)
- English language and style
R1 suggest minor spell check.
This has now been done.
General comments
R1 suggest that The Introduction can be improved. Reviewer 2 also suggested this.
Under R2 are given the specific changes made.
- Specific comments
- L21-22: R1: The sentence in lines 21 and 22 is non informative (‘Both positive and negative impact on nutrient content of the substitutions were found, but only few of nutritional importance. ‘). It must be more specific or be deleted.
Answer: The sentence has been rephrased and specified as suggested:
‘As regards nutrients both positive and negative impact of the substitution were found, but only few of particular nutritional importance, indicating that replacing beef with a combination of other foods and maintaining or even improving the nutrient profile of the diet is possible.’
- R1: The Danish population is based on a relatively old study from 2005-2008, more than 10 years ago. More recent data would be more useful; for instance, EFSA has probably recent studies available.
Answer: It is true that food intake data are from 2005-2008, which are included in the EFSA database. Although newer Danish data (2011-13) exist (however not available through the EFSA database) the data from 2005-2008 were used since the dietary patterns were available from this older dietary survey, and since we believe they can serve as examples of real-life dietary patterns. No changes has been made.
- R1: Table 1: why such large difference in GHG between Heifer and Cow?
Answer: The explanation for this has been added in Table 1 – see note 1.
- When allocating the environmental impact of the dairy system, milk is the main product that pays the major part of the environmental cost of the system and the co-products, meat from the dairy cow sent to slaughter and the new born calf is only responsible for the marginal cost of their production (see Mogensen et al. [22]). Likewise, in beef breed production represented by Limousine (Intensive) and Scottish Highland (extensive) the weaned calves raised for slaughter are the main product that is responsible for the major part of the environmental cost of the system, whereas the co-product, meat from the cow sent to slaughter, only pays the marginal cost for its production (see Mogensen et al. [22]).
Reviewer 2 Report
OVERALL:
I think the paper is interesting and the fact that you considered different existing diet patterns is quite positive.
Please go back and use an English language program or native speaker to check the language. There seem to be a number of missing “articles” – a, an, the – that need to be added. Most English language grammar check programs would catch this. There is a little bit of stilted language that a native speaker could advise on, but generally it is fine.
The title says “The environmental …”, but the main point of the article related to environmental is GHG with a little bit of info energy use. But that information seems limited to “on farm” land use and does not account for such things as water conservation, etc. Thus, I don’t really think that the title represents “environmental” as much as it does a limited set of environmental factors. Perhaps “Selected environmental …” should be used in the title
One thing I find a bit troubling is that there appears to be a little bit of a slant toward showing the downside of beef without showing the positive side, particularly in the introduction. For example, there is nothing in the introduction about the primary reasons people eat beef and other meats – liking. Consumers like the flavor and texture and eating beef gives them pleasure. Nutrition, health, environmental concerns are all lower in importance than liking for most consumers. That is not to say they are not important and should not be considerations, but as the primary factor in food choice, liking should at least be mentioned.
My take-away from the data in this paper is that any dietary pattern can be constructed to go higher or lower in GHG and that ultimately, if that is a consideration, GHG may be reduced by a few percentages. Is that correct? If so, that needs to be more forcefully stated in a more concise way.
Specific comments:
ABSTRACT:
“The Green diet had (the) same CF as the other 18 diets, despite having the lowest intake of beef as well as the lowest intake of red meat in total. Substituting beef with other animal products or legumes, reduced the diets’ (which diets) CF by 4-12 % and LU 20 by 5-14 %.” These two sentences seem to contradict each other. Note the comments in parentheses for missing words or needed explanations in addition to the apparent contradiction.
“Both positive and negative impact on nutrient content of the substitutions were found, but only (a) few of nutritional importance. A combination of substitute foods may maintain or improve the nutritional profile.” What does this really tell us? Isn’t this true of ANY diet combination. I can create a diet adding or deleting ANY food and accomplish this if I try? Why is this unique or special to this study and is it really something unique to taking beef out of the diet?
PAPER:
Water use by people in the so-called “green” diet is much higher than any other diet. Total liquid intake also seems to be much higher. What impact does this have on the environmental impact, including GHG of using so much water – e.g. sanitation of water both before use and after use? Do we have any idea? Seems like this should at least be noted in the results or discussion as a point of future study.
Line 496: “Replacing 50% of total meat in the Swedish average diet with domestically (itlaics added) grown grain legumes.” Is domestically really possible? Are there enough legumes grown in Nordic countries to replace beef? I really question the use of such statistics if this is not realistic. If it is not, then imported products must be used, which raising the environmental footprint because of transportation issues. It seems that this should at least be mentioned.
The paper seems a bit long and wanders a bit to somewhat tangential topics (such a 4.4 on sugars – why sugars and not all contributions on diet?). Is all of this really needed? At times I thought perhaps you were going to show something quite new and different in the analysis and you came back to the same basic result – not much difference in environmental impact. Thus, I wonder if all the additional analysis and verbiage is to sort of soften the blow that you didn’t find what you expected or hoped to find. Please make the paper more concise and focus on what you really want to say. I think you have a nice paper, but frankly I got a bit bored reading so much, when you kept coming back to the same conclusion. It reads more like a dissertation or thesis where you are attempting to defend every piece of data rather than a published research paper with an overarching theme that addressed a concise topic. I appreciate the depth, but I also appreciate it being said in fewer words.
Line 562. “in the future dietary guidelines will have to be reframed to incorporate environmental and climate considerations.” That is a strong statement and assumes that dietary recommendations are written for quite a broad purpose. As they stand now, dietary recommendations in almost every country are based on nutritional needs and health – period. They do not include other issues such as liking and pleasure, economics, convenience, environmental, etc. I am not suggesting that dietary guidelines should not include those, but if they should (as you state in 562), then you should take a broader view and suggest environmental concerns be added as part of a reconsideration of dietary guidelines to include more issues than just nutrition/health.
The conclusions are much more "on target" than the results and discussion. They seem more focused and more "on-topic" with just a little bit of the last part being a bit unfocused since it comes back to the any diet can do something phrase.
Author Response
Dear Editor and reviewer 2,
Thank you very much for the helpful comments, suggestions and questions from you.
We have now revised the paper according to these and have been able to change almost everything in the suggested way as can be seen in the List of changes.
Looking forward to hear further decisions about this paper.
Yours sincerely,
Lisbeth Mogensen
Authors reply to the Review Report
- Reviewer 2 (R2)
- English language and style
R2 suggest minor spell check. ‘Please go back and use an English language program or native speaker to check the language. There seem to be a number of missing “articles” – a, an, the – that need to be added. There is a little bit of stilted language that a native speaker could advise on, but generally it is fine.’
Answer: This has now been done.
- Overall comments
- R2: The title says “The environmental …”, but the main point of the article related to environmental is GHG with a little bit of info energy use. But that information seems limited to “on farm” land use and does not account for such things as water conservation, etc. Thus, I don’t really think that the title represents “environmental” as much as it does a limited set of environmental factors. Perhaps “Selected environmental …” should be used in the title
Answer:
We agree on this and have changed the title from ‘environmental impact’ to ‘climate impact’.
- R2: One thing I find a bit troubling is that there appears to be a little bit of a slant toward showing the downside of beef without showing the positive side, particularly in the introduction. For example, there is nothing in the introduction about the primary reasons people eat beef and other meats – liking. Consumers like the flavor and texture and eating beef gives them pleasure. Nutrition, health, environmental concerns are all lower in importance than liking for most consumers. That is not to say they are not important and should not be considerations, but as the primary factor in food choice, liking should at least be mentioned.
Answer: This has been taken into account by adding the following section as the very beginning of the Introduction:
‘Red meat, including beef products, plays a central role in Western diets due to taste preferences, culinary tradition and social norms and has been recognized as an important provider of nutrients. Thus, consumption of meat is typically high in most developed countries. However, it is well recognized that the production of meat and especially beef products is associated with a high load of greenhouse gas (GHG) emissions ……’
- R2: My take-away from the data in this paper is that any dietary pattern can be constructed to go higher or lower in GHG and that ultimately, if that is a consideration, GHG may be reduced by a few percentages. Is that correct? If so, that needs to be more forcefully stated in a more concise way.
Answer: It is true that the Traditional, Fast-food and Green dietary pattern showed almost the same GHG emission per person per day, however the High-beef diet showed an 11% higher GHG emission than the Average diet. To clarify that, we have added the found GHG levels of the diet in the Abstract:
‘….Beef products accounted for 513, 560, 409, 1023 g CO2eq per day, respectively, in the four dietary patterns (Traditional, Fast-food, Green, and High-beef). CF of the total diets were 4.4, 4.2, 4.3 and 5.0 kg CO2eq per day (10 MJ), respectively…’
As well as the effects of substituting beef with other food products has been clarified in the Abstract:
‘The Green diet had same CF as the Traditional and the Fast-food diets, despite having the lowest intake of beef as well as the lowest intake of red meat in total. A theoretical substitution of beef with other animal products or legumes in each of these three diets, reduced the diets’ CF by 4-12 % and LU by 5-14 %’
- Specific comments
ABSTRACT:
- R2: “The Green diet had (the) same CF as the other 18 diets, despite having the lowest intake of beef as well as the lowest intake of red meat in total. Substituting beef with other animal products or legumes, reduced the diets’ (which diets) CF by 4-12 % and LU 20 by 5-14 %.” These two sentences seem to contradict each other. Note the comments in parentheses for missing words or needed explanations in addition to the apparent contradiction.
Answer: Has been rephrased to consider this:
‘The Green diet had almost the same CF as the Traditional and the Fast-food diets despite having the lowest intake of beef as well as the lowest intake of red meat in total. A theoretical substitution of beef with other animal products or legumes in each of these three diets, reduced the diets’ CF by 4-12 % and LU by 5-14 % ’
- R2: L21-23: “Both positive and negative impact on nutrient content of the substitutions were found, but only (a) few of nutritional importance. A combination of substitute foods may maintain or improve the nutritional profile.”
What does this really tell us? Isn’t this true of ANY diet combination. I can create a diet adding or deleting ANY food and accomplish this if I try? Why is this unique or special to this study and is it really, something unique to taking beef out of the diet?
Answer: This sentence has been rephrased to clarify the findings:
‘As regards nutrients both positive and negative impact of the substitution were found, but only few of particular nutritional importance, indicating that replacing beef with a combination of other foods and maintaining or even improving the nutrient profile of the diet is possible .’
PAPER:
- R2: Water use by people in the so-called “green” diet is much higher than any other diet. Total liquid intake also seems to be much higher. What impact does this have on the environmental impact, including GHG of using so much water – e.g. sanitation of water both before use and after use? Do we have any idea? Seems like this should at least be noted in the results or discussion as a point of future study.
Answer: We did take into account the GHG related to the different types of liquid intake in the different diets, including tap- water and bottle water. As regards other environmental impacts, we refrained from including water foot printing in this study, since sufficient data are far from being available. What we do know is that the so-called blue water footprint (consumptive water) is high for meat products compared to plant based foods. E.g., we have shown a water consumption of 3-6 l per 100 g of edible pork product (Raffn et al 2019. J Cleaner Prod 233:1355-1365). For beef products, higher values are proposed (Ridoutt et al 2012. Int J. Life Cycle Assess 17:165-175. Thus, there is no reason to believe that that water use in the Green diet should represent a particular problem compared to the other diets.
No changes were made in the text.
- R2: Line 496: “Replacing 50% of total meat in the Swedish average diet with domestically (italics added) grown grain legumes.” Is domestically really possible? Are there enough legumes grown in Nordic countries to replace beef? I really question the use of such statistics if this is not realistic. If it is not, then imported products must be used, which raising the environmental footprint because of transportation issues. It seems that this should at least be mentioned.
Answer:
L 496: ‘domestically grown’ has been deleted from the sentence as it is not an important issue for the nutritional discussion here whether the legumes are grown domestically or imported.
However, in the paper by Röös et al. (2018) [51] they found, that replacing 50% of total meat in the Swedish average diet with domestically grown grain legumes will reduce land requirement by 23%. There would therefore be a net surplus of approx. 21,500 ha land in Sweden and more than enough cropland is made available in the transition scenario by reduced cereal and rapeseed cultivation (less animal feed),for expanding grain legume cultivation to the required level.
This is another – but important discussion – but it will not be included in this paper.
- R2: The paper seems a bit long and wanders a bit to somewhat tangential topics (such a 4.4 on sugars – why sugars and not all contributions on diet?). Is all of this really needed? At times I thought perhaps you were going to show something quite new and different in the analysis and you came back to the same basic result – not much difference in environmental impact. Thus, I wonder if all the additional analysis and verbiage is to sort of soften the blow that you didn’t find what you expected or hoped to find. Please make the paper more concise and focus on what you really want to say. I think you have a nice paper, but frankly I got a bit bored reading so much, when you kept coming back to the same conclusion. It reads more like a dissertation or thesis where you are attempting to defend every piece of data rather than a published research paper with an overarching theme that addressed a concise topic. I appreciate the depth, but I also appreciate it being said in fewer words.
Answer:
Section 4.4 has been deleted as suggested by R2. However, the main messages about 1) uncertainty on carbon footprint on sweets and beverages and 2) that reduced intake of sweet and beverages could be an important mitigation option has been moved to the new section 4.4 and 4.5 respectively.
Text moved to 4.4:
‘Especially, when it comes to the more processed foods like some sweets and beverages, the number of LCA studies behind these food groups are much lower and the uncertainty for these GHG estimates per item similarly higher.’
Text moved to 4.5:
‘In the present study, sugar and sweets and beverages contribute with 27% of the GHG emissions of the Average diet. The relatively high intake per day gives this substantial contribution to the GHG emission of the diet. Saxe et al. [69] and Moberg et al. [70] found similar GHG emissions caused by the consumption of sweets, snacks and drinks (excluding milk). At the same time, a high intake of soft drinks, sweets, and alcohol is negatively associated to health [72-75] and both NNR2012 [40] and WHO suggest to lower the intake hereof [76]. ‘
Besides that, there has been other reductions in the text in the Discussion.
- R2: Line 562. “in the future dietary guidelines will have to be reframed to incorporate environmental and climate considerations.” That is a strong statement and assumes that dietary recommendations are written for quite a broad purpose. As they stand now, dietary recommendations in almost every country are based on nutritional needs and health – period. They do not include other issues such as liking and pleasure, economics, convenience, environmental, etc. I am not suggesting that dietary guidelines should not include those, but if they should (as you state in 562), then you should take a broader view and suggest environmental concerns be added as part of a reconsideration of dietary guidelines to include more issues than just nutrition/health.
Answer: You are of course right FBDG in all countries are primarily based on nutritional needs and health, although both FAO/WHO and EFSA giudelines for developing FBDG also include cultural acceptability, accessibility etc. (“Food-based dietary guidelines constitute science-based policy recommendations in the form of guidelines for healthy eating. They are primarily intended for consumer information and education, and as such, they should be appropriate for the region or country, culturally acceptable and practical to implement. Moreover, they should be consistent, easily understood and easily memorable.” (EFSA Panel on Dietetic Products, Nutrition, and Allergies (NDA); Scientific Opinion on establishing Food-Based Dietary Guidelines. EFSA Journal 2010; 8(3):1460. [42 pp.]. doi:10.2903/j.efsa.2010.1460. Available online: www.efsa.europa.eu))
A paper from FAO (2012) defined sustainable diets by including both health and environmental issues and much more: “Sustainable Diets are those diets with low environmental impacts which contribute to food and nutrition security and to healthy life for present and future generations. Sustainable diets are protective and respectful of biodiversity and ecosystems, culturally acceptable, accessible, economically fair and affordable; nutritionally adequate, safe and healthy; while optimizing natural and human resources.” (Barbara Burlingame and Sandro Dernini (eds.): SUSTAINABLE DIETS AND BIODIVERSITY DIRECTIONS AND SOLUTIONS FOR POLICY, RESEARCH AND ACTION. FAO 2012).
More recently (2019) FAO/WHO has developed guiding principles for sustainable healthy diets which “are food based, and take into account nutrient recommendations while considering environmental, social/cultural and economic sustainability”.
We have changed the wording and added a sentence about this:
‘Therefore in order to take environmental and climate into account future dietary guidelines should be reframed. A few countries have done this recently, and FAO/WHO has in 2019 developed The Guiding Principles for Sustainable Healthy Diets which “are food based, and take into account nutrient recommendations while considering environmental, social/cultural and economic sustainability” and they encouraged countries to start the process of implementation of sustainable healthy guidelines.’
- R2: The conclusions are much more "on target" than the results and discussion. They seem more focused and more "on-topic" with just a little bit of the last part being a bit unfocused since it comes back to the any diet can do something phrase.
Answer: The last part of the conclusion has been rephrased to consider this:
‘ The total High-beef diet had the highest GHG emissions of all the dietary patterns while the Green and the Fast-food diets were only slightly lower than the Traditional diet and the Average diet. The substitution of beef with other protein rich foods showed this as one possible way to obtain lower GHG emissions from the diet by 4-12 % and the land use by 5-14 % and at the same time keep or even improve the nutritional quality of the diet.’
Round 2
Reviewer 2 Report
the authors made some attempt to answer my concerns, but many of them remain.
I am disappointed that the authors did not carefully review the paper for grammatical and English errors. In fact, they actually increased the errors (e.g. in a change to the abstract they left out a "." after the sentence ending in "LU by 5-14%". There are other issues similar to that in the paper that have not been corrected. For example, abbreviations in the tables are used but are not defined in the table - Tables should stand alone. In addition words and phrases that make no sense in English are used (e.g. "recommend on evidence regarding the risk of" - line 461 - do you mean "recommend based on a risk assessment for"s ). There are many other examples (line 470 "Cooked dry brown beans was used...", the subject and verb do not agree - beans are plural, the verb is singular. Cooked dry beans WERE used... "
The grammar and wording must be checked carefully.
I commented before tht the "slant" of the paper to that of "beef is not a particularly good option" was inappropriate and the authors made some minor changes. however, it is still quite evident in the paper, yet the data do not necessarily support that from the standpoint of GHG, except only marginally - there may a statistical difference, but is the difference relevant and practical. The authors comment that the difference in GHG for the traditional and fast food dietary patterns is not much different from the Green option and that, in fact, the GHG for the High Beef Diet is not much higher either. They go on to show that the decrease in GHG is 4-12% when various dietary changes are made. Yet they do not explain what that means.
First, that is a fairly low percentage. Second, if my calculations are correct, the savings over a year in GHG equates to changing from 1 (one) incandescent light bulb to an energy-efficient light bulb for a year (for the lowest percentage decrease) to driving 2.6km less per day for a year (for the highest percentage decrease). That is only a marginal difference when one considers the difficulty in changing eating patterns, but none of that is mentioned. In fact, the authors go on to say when that "The rather simple substitutions in the present study... (line 453)", which makes this seem like an easy modification. It was easy for the authors to do, but switching from eating a high beef diet to eating legumes as a replacement for beef (the assumption in one scenario) would be incredibly difficult for the average citizen to do. That is not mentioned - it must be!
I noted previously that the bias toward less beef in the diet starts in the abstract and continues on throughout the paper with little mention of the fact that people eat beef because they LIKE it. Liking is the single most important criteria for food consumption - environmental concerns are way down the list of importance. Such information is clearly mentioned in papers on food choice such as those by Phan and Chambers (Journal of Sensory Studies; Appetite, both published in 2016) and others.
Suggesting that foods that are liked should be replaced by foods that produce a slightly more environmentally friendly diet is completely unrealistic. Yet that appears to come through in this paper as the main suggested recommendation. Even though you don't specifically make that statement, it is implied through your abstract, introduction, results, discussion, and conclusions and makes the entire discussion meaningless. This must be changed. You MUST explain the relevance of this data in terms of realistic expectations for change. A 4% change is an interesting phenomenon but is unlikely to change many people's perspective in terms of actually changing eating behavior. It is a very small change in a quite small part of overall GHG production.
I have some concerns that you make statements about so-called "positive" and "negative" foods, some of which are controversial and imply that specific foods are good or bad, which we know to be incorrect. All foods can be appropriate if included in reasonable amounts and some foods are included too often. People speak of foods like soft drinks as bad, but they are only bad if too much is consumed to the detriment of other foods. The same is true for carrots - if you eat a kg of carrots every day they are bad for you - too much Vit. A and you will get too few other nutrients. A great case in point in your paper is talking about potatoes as "positive". There are nutritionists who view potatoes as really bad since they have starch that seems to break down fast and enter the bloodstream rapidly causing a spike in blood sugar levels. I don't think the positive and negative adjectives are necessary (cimply leave them out of the sentences and they still make sense for the most part).
They also create a problem when you get to the dietary data. You modify the diets, but do not do so on the basis of dietary guidelines because you have these positive and negative foods. You tend to increase potatoes (but there is no dietary guideline for that) and rye bread (again no dietary guideline for rye bread specifically). Then you lump fresh-cooked beef and processed beef together, which has a huge impact on your sodium levels since fresh cooked beef is not a high source of sodium. The dietary information is interesting, but you place too much weight on it to boost your statement that dietary changes and GHG seem to go hand in hand. The fact is that you chose some easy to calculate generic dietary changes (OK I understand that and am happy that you did it at all), but instead of suggesting that dietary change and GHG seem to go together, treat it for what it is - an interesting extra piece of data from easy to calculate generic dietary changes. Then state that such information is interesting and shows what could be possible IF people could make such changes, but also show that this is a limitation in terms of both behavior change (unlikely to get people to do this easily) and these represent single substitution changes that are unrealistic in such a complex system but do illustrate the complexity of the issue.
The paper needs to have a more balanced discussion and conclusions that do not imply that an easy dietary change (no it is not) would appear to have a bigger change in GHG production than it actually does. The power of your data is in showing that 1) diet is one aspect off GHG production, 2) that ALL diets contribute to GHG production, 3) that some diets contribute SLIGHTLY more than others - a surprise for many people who believe only meat production affects GHG, 4) that changes to the dietary pattern are complex and affect more than just GHG production (e.g. nutritional composition, which does change and will need to be watched for ALL groups). The power of your data is not the last sentence of your abstract which seems to imply much more than you could and should conclude from this paper.
Author Response
Cover letter – Response to Reviewer 2 (R2) - 2. Review
Manuscript Foods-879656
English language and style
R2:
I am disappointed that the authors did not carefully review the paper for grammatical and English errors. In fact, they actually increased the errors (e.g. in a change to the abstract they left out a "." after the sentence ending in "LU by 5-14%". There are other issues similar to that in the paper that have not been corrected. In addition words and phrases that make no sense in English are used (e.g. "recommend on evidence regarding the risk of" - line 461 - do you mean "recommend based on a risk assessment for"s ). There are many other examples (line 470 "Cooked dry brown beans was used...", the subject and verb do not agree - beans are plural, the verb is singular. Cooked dry beans WERE used... " The grammar and wording must be checked carefully.
Answer:
The grammar and wording has been checked carefully by a professional English editing service, and it is now implemented.
R2:
For example, abbreviations in the tables are used but are not defined in the table - Tables should stand alone.
Answer:
Abbreviations in the tables has now been defined in the tables.
R2:
I commented before tht the "slant" of the paper to that of "beef is not a particularly good option" was inappropriate and the authors made some minor changes. However, it is still quite evident in the paper, yet the data do not necessarily support that from the standpoint of GHG, except only marginally - there may a statistical difference, but is the difference relevant and practical. The authors comment that the difference in GHG for the traditional and fast food dietary patterns is not much different from the Green option and that, in fact, the GHG for the High Beef Diet is not much higher either. They go on to show that the decrease in GHG is 4-12% when various dietary changes are made. Yet they do not explain what that means.
Answer:
We think it is important to distinguish between beef per se and a diet that includes beef. It is well recognized that beef in itself ‘is not a god option as regards GHG’, but what was not clear from the literature, was how beef is included in different dietary patterns and what this means for the GHG emission of the total diet. (How different segments adjust for differences in beef intake, and what this means for the GHG emission of the total diet). And yes, we find that there are only minor differences between three of the diets despite differences in beef intake, whereas the high beef diet stands out with a higher GHG emission. Despite the fact that there are only small differences found in the GHG emission of the total diets (except the high beef), it is still relevant to consider how GHG emission could be reduced by replacing beef with other food items, that is why we investigate the consequences of ‘simple’ dietary changes. We do not agree that a difference between diets of 16 % (high beef versus Green) is a minor thing given the fact that food intake in general is expected to be responsible for ¼ of total GHG emissions. So, we think that it is justified that the ‘tone’ of the paper is that the intake of beef is a concern as regards GHG.
We agree that it could be clearer in the paper and we have made the following amendments:
L30-31: ‘in particularly compared to plant based foods but…’ has been added to the sentence:
‘However, it is well recognised that the production of meat and especially beef products is associated with a high load of greenhouse gas (GHG) emissions, in particularly compared to plant based foods but also compared to other types of meat [1, 2, 3].’
L36-39: We have rephrased/added the sentence:
‘There is an increasing demand of reducing the total climate impact of our diets as it has been estimated that todays’ food supply is responsible for 26% of the anthropogenic GHG emission [3]. Therefore, there is an increasing focus on how dietary changes can reduce climate impact [5, 6, 7].’
R2: They go on to show that the decrease in GHG is 4-12% when various dietary changes are made. Yet they do not explain what that means.
First, that is a fairly low percentage. Second, if my calculations are correct, the savings over a year in GHG equates to changing from 1 (one) incandescent light bulb to an energy-efficient light bulb for a year (for the lowest percentage decrease) to driving 2.6km less per day for a year (for the highest percentage decrease). That is only a marginal difference when one considers the difficulty in changing eating patterns, but none of that is mentioned.
Answer:
We think your calculations are right about relating the saved GHG emissions from food intake to car driving. However, we have chosen to focus section 4.5 in the discussion on the mitigation options that relate to food intake.
The maximum potential for reduction in GHG emission from food by dietary changes has now been quantified
L609-610: ‘The CF of the Green diet or the Fast-food diet was around 16-19% lower than the CF of the High-beef diet.
L625-628: ‘For further GHG reduction, a reduction of all animal based food could be considered as exclusion of animal-based products from the current diets has been found to have a huge potential to reduce land use by up to 76% and GHG emissions by up to 55% [3, 5, 16, 17, 78].’
The idea by adding these levels of mitigation potential, is that the reader can relate our findings of a decrease in GHG of 4-12% to the maximum potential for mitigation.
For the same reason, so the reader can relate our found mitigation potential to other possible mitigations, the mitigation potential in reducing intake of soft drinks, sweets and alcohol was included in the discussion.
To the section L 631-641: ‘In the present study, sugar, sweets and beverages (excluding milk and juice) contribute with 27% of the GHG emissions of the Average diet. ….’
We have added L637-641 to that discussion:
‘These foods are often related to excessive energy intake and a reduction of CF of the diet by up to 10% has been estimated if energy intake was balanced to energy expenditure [16]. To keep the energy level of the dietary patterns at 10 MJ substitution of sweets and beverages with other foods as whole grain, fruits and vegetables is needed and will affect the reduction of the CF of the diet.’
L660-667: We have added and changed the wording in the conclusion: ‘The total High-beef diet had the highest GHG emissions of all the dietary patterns, 16-19% higher than the Green and the Fast-food diets, which were only slightly lower than the Traditional diet and the Average diet, since higher intake of other foods than beef contributed to the total GHG emissions from the diets. In addition, a substitution of beef with other protein rich foods showed this is one possible way to obtain lower GHG emissions from the three identified Danish dietary patterns (Traditional, Fast-food and Green) by up to 12 %, highest for substitution with egg and legumes. At the same time it seems possible to keep or even improve the nutritional quality of the diets.’
And we have deleted a sentence L651-653.
R2: In fact, the authors go on to say when that "The rather simple substitutions in the present study... (line 453)", which makes this seem like an easy modification. It was easy for the authors to do, but switching from eating a high beef diet to eating legumes as a replacement for beef (the assumption in one scenario) would be incredibly difficult for the average citizen to do. That is not mentioned - it must be!
Answer:
We agree that dietary changes are challenging, and people need different kind of support to change their diet, if possible. In the paper we refer to the simplicity of the substitution in the study since we have substituted beef whit other foods one by one, not with different combinations of foods. The aim of the paper was to investigate the possible climate impact and nutritional impact of different dietary patterns and dietary changes. We do not want to indicate that it is simple for people to make these changes.
To make this clearer in the paper we have made the following amendments/additions:
L77-79: We have added ‘… as well as theoretical estimations of impact of further changes’ to the sentence:
‘These observations highlight the importance of considering real-life diets as a basis for evaluating the impact of diets on GHG emissions and nutritional quality simultaneously, as well as theoretical estimations of impact of further changes.’
L91-96: To the objective we have added: ‘as well as the potential impact of replacing beef with other foods in these patterns.’
‘The objective of the present work was therefore to estimate the GHG emission and land use of the beef products consumed in Denmark, also taking into account the resource use and losses in the chain from the slaughterhouse until the beef product is ready to eat, and to use these new data to investigate the carbon footprint and land use as well as the nutritional profile of the total diet of different Danish dietary patterns as well as the potential impact of replacing beef with other foods in these patterns. ‘
L 241: As an introduction to section 2.5 we have added: ‘In order to estimate the potential climate impact of excluding beef from the diets,…’
L244: We have added: ‘of pork, poultry, fish, eggs or cheese’
L410-414: We have changed the wording to:
‘Any suggested dietary change in order to reduce the environmental impact of our diets should take into account the impact on the nutritive value of the diet. Due to the fact that the GHG emission per kg beef or per MJ beef is much higher than for most other regular foods, the potential impact of replacing beef with other foods in these patterns was investigated’
L481-485: we have added this introductory remark to section 4.2_
‘Dietary changes in accordance with FBDG are promoted in countries all over the world in order to improve health and reduce the risk of chronic diseases (http://www.fao.org/nutrition/nutrition-education/food-dietary-guidelines/en/). The increasing focus on dietary changes based on the increasing demand for reducing the climate impact of the diet provides the opportunity of dietary changes that at the same time might fulfil the FBDG and improve the nutritional quality and health effect of the diet. ‘
486-488: we have added this explanation of simple substitution:
‘In the present study we estimated the potential reduction of the CF of the diet by rather simple substitutions where we calculate the effect of replacing beef with one other food at the time.’
R2:
I noted previously that the bias toward less beef in the diet starts in the abstract and continues on throughout the paper with little mention of the fact that people eat beef because they LIKE it. Liking is the single most important criteria for food consumption - environmental concerns are way down the list of importance. Such information is clearly mentioned in papers on food choice such as those by Phan and Chambers (Journal of Sensory Studies; Appetite, both published in 2016) and others.
Suggesting that foods that are liked should be replaced by foods that produce a slightly more environmentally friendly diet is completely unrealistic. Yet that appears to come through in this paper as the main suggested recommendation. Even though you don't specifically make that statement, it is implied through your abstract, introduction, results, discussion, and conclusions and makes the entire discussion meaningless. This must be changed. You MUST explain the relevance of this data in terms of realistic expectations for change. A 4% change is an interesting phenomenon but is unlikely to change many people's perspective in terms of actually changing eating behavior. It is a very small change in a quite small part of overall GHG production.
Answer:
We have already argued the relevance of food intake as contributor to the the anthropogenic GHG emission (L36-42). The aim of the paper was to investigate the possible climate impact and nutritional impact of different dietary patterns and dietary changes. We don’t want to get into how these changes should be implemented, since this is out of scope of this paper. We agree it is relevant, but we also realize that this is a huge topic and a scientific area different from ours.
We also agree that this could be clearer in the paper, and we have made the following amendments
L603-607 We have added:
‘Recognizing that other issues as liking, pleasure, convenience and price are strong motivators of dietary choices [112], the present study is limited to investigate the potential impact on CF and nutritional value of the diets in order to be able to suggest changes that take both the need of CF reduction and nutritional improvements into account. How to promote such changes effectively is challenging and not within the scope of this paper.’
L375-377 We have exchanged the text:
“Any guidance on ways to reduce the environmental impact of our diets needs to take into account the impact on the nutritive value of the diet.”
with
“In addition to the comparison of the climate impact of the diets comparison of the nutritional quality of the diets is highly relevant.”
R2:
I have some concerns that you make statements about so-called "positive" and "negative" foods, some of which are controversial and imply that specific foods are good or bad, which we know to be incorrect. All foods can be appropriate if included in reasonable amounts and some foods are included too often. People speak of foods like soft drinks as bad, but they are only bad if too much is consumed to the detriment of other foods. The same is true for carrots - if you eat a kg of carrots every day they are bad for you - too much Vit. A and you will get too few other nutrients. A great case in point in your paper is talking about potatoes as "positive". There are nutritionists who view potatoes as really bad since they have starch that seems to break down fast and enter the bloodstream rapidly causing a spike in blood sugar levels. I don't think the positive and negative adjectives are necessary (cimply leave them out of the sentences and they still make sense for the most part).
They also create a problem when you get to the dietary data. You modify the diets, but do not do so on the basis of dietary guidelines because you have these positive and negative foods. You tend to increase potatoes (but there is no dietary guideline for that) and rye bread (again no dietary guideline for rye bread specifically). Then you lump fresh-cooked beef and processed beef together, which has a huge impact on your sodium levels since fresh cooked beef is not a high source of sodium.
Answer:
We agree that it is incorrect to describe foods as either positive or negative, and we didn’t do so. We described the increase or decrease of the content of different foods to be on the positive or negative side. Since this was done in relation to the Danish FBDG and this seems unclear, we have made the following amendments:
L329-336: we have added:
‘Compared to the Danish FBDG, which are health based, the Danish average adult diet would improve by increase of the content of fruit and vegetables, whole grain, fish and fats from vegetable sources (except coconut fat and palm oils) instead of animal source; and a decrease in red meat (beef, pork and lamb), alcohol containing beverages and sugar containing food and beverages [41, 48]. According to the Danish FBDG potatoes are a positive part of a healthy diet, referring to boiled potatoes which are common in the Danish dietary culture – and not French fries. Rye bread and oat meal are also common in the Danish diet and important contributors of whole grain, while wheat bread constitute both white and whole grain types of bread [41, 48].’
And in the text L337-349 we have deleted: “on the positive side” and “on the negative side”
R2:
The dietary information is interesting, but you place too much weight on it to boost your statement that dietary changes and GHG seem to go hand in hand. The fact is that you chose some easy to calculate generic dietary changes (OK I understand that and am happy that you did it at all), but instead of suggesting that dietary change and GHG seem to go together, treat it for what it is - an interesting extra piece of data from easy to calculate generic dietary changes. Then state that such information is interesting and shows what could be possible IF people could make such changes, but also show that this is a limitation in terms of both behavior change (unlikely to get people to do this easily) and these represent single substitution changes that are unrealistic in such a complex system but do illustrate the complexity of the issue.
Answer:
Our data shows that GHG and healthy dietary profile do not automatically go hand in hand as illustrated and discussed in section 4.3: The ‘un-expected’ high GHG emissions from food in the Green diet. As regards dietary changes we show the potential impact but we have now highlighted that implementation of these changes in real life diets is demanding.
L603-607 we have added:
‘Recognizing that other issues as liking, pleasure, convenience and price are strong motivators of dietary choices [112], the present study is limited to investigate the potential impact on CF and nutritional value of the diets in order to be able to suggest changes that take both the need of CF reduction and nutritional improvements into account. How to promote such changes effectively is challenging and not within the scope of this paper.’
L667-669 in the Conclusion: we have added:
‘However, additionally dietary changes in the direction of a more plant based diet are needed for a larger climate impact, but such changes of dietary habits are even more challenging than the beef reduction, and the changes need additional attention to be sure that the diets are healthy and sufficient.’
R2:
The paper needs to have a more balanced discussion and conclusions that do not imply that an easy dietary change (no it is not) would appear to have a bigger change in GHG production than it actually does. The power of your data is in showing that 1) diet is one aspect off GHG production, 2) that ALL diets contribute to GHG production, 3) that some diets contribute SLIGHTLY more than others - a surprise for many people who believe only meat production affects GHG, 4) that changes to the dietary pattern are complex and affect more than just GHG production (e.g. nutritional composition, which does change and will need to be watched for ALL groups).
Answer:
We think that the above mentioned changes have addressed this point.
R2:
The power of your data is not the last sentence of your abstract which seems to imply much more than you could and should conclude from this paper.
Answer:
We have rephrased the last sentence of the Abstract, L19-22:
‘As regards nutrients both positive and negative impacts of these substitutions were found but only a few of particular nutritional importance, indicating that replacing beef with a combination of other foods without significant effect on the nutrient profile of the diet is a potential mitigation option.’
L448-449 We have deleted similar text.
Further, to make the text clearer we have included the following amendments:
L 357-358: Deleted since it is included in the discussion
L415-416: We have added text and changed the wording to: the critical nutrients (which were identified in relation to nutritional evaluation of the different diets, table 6)
L468: ‘effect is’ is changed to ‘differences are’
L516-520 we have moved the following text up from the end of the section since we here change focus from the substitutions to the different diet:
‘The nutritional quality of the four dietary patterns was evaluated both in relation to the content of foods and food groups related to the Danish FBDG and with regards to the nutrient content. The distribution of nutrient intakes of the average diet of adults were used to point out nutrients of concern for inadequacy. As expected the average Green diet, which was closer to the FBDG than the other diets, was also closer to the nutrient recommendations. ‘
L546-548: We have deleted a sentence, since it is mentioned in the method section
L555: we have deleted some words (‘Of the different diets’) in the beginning of the sentence
L566-567: We have added some words to: ‘These changes in the diets where the red meat was reduced but total meat content was approximately unchanged’